# Impact of ASOS Real-Time Quality Control on Convective Gust Extremes in the USA

Nicholas John Cook 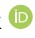

Independent Researcher, Highcliffe-on-Sea, Dorset BH23 5DH, UK; wind@njcook.uk

**Abstract:** Most damage to buildings across the contiguous United States, in terms of number and total cost, is caused by gusts in convective events associated with thunderstorms. Their assessment relies on the integrity of meteorological observations. This study examines the impact on risk due to valid gust observations culled erroneously by the real-time quality control algorithm of the US Automated Surface Observation System (ASOS) after 2013. ASOS data before 2014 are used to simulate the effect of this algorithm at 450 well-exposed stations distributed across the contiguous USA. The peak gust is culled in around 10% of these events causing significant underestimates of extreme gusts. The full ASOS record, 2000–2021, is used to estimate and map the 50-year mean recurrence interval (MRI) gust speeds, the conventional metric for structural design. It is concluded that recovery of erroneously culled observations is not possible, so the only practical option to eliminate underestimation is to ensure that the 50-year MRI gust speed at any given station is not less than the mean for nearby surrounding stations. This also affects stations where values are legitimately lower than their neighbors, which represents the price that must be paid to eliminate unacceptable risk.

**Keywords:** thunderstorm; extreme gust speed; quality control; extreme value analysis; ASOS; convective gust events

## 1. Introduction

The risk of damage by wind action on buildings and structures around the world is mitigated by regulations, codes, and standards that specify minimum design wind speeds to be used in design calculations. These design codes regulate the design of all manner of structures in a way that balances risk to life against economic cost of construction and repair. Most codes specify peak wind speeds with a datum mean recurrence interval (MRI), which ranges from 50 years for buildings to 1000 years for critical infrastructures, that is determined by extreme-value analysis of peak wind gusts and is therefore totally reliant on the integrity of meteorological observations.

Until recently, the codes have assumed that the design wind gust originates from turbulence in the atmospheric boundary layer (ABL) generated by synoptic-scale windstorms. However, most damage to buildings across the contiguous United States (CONUS), in terms of number and total cost, is now known to be caused by gusts from thunderstorm downbursts [1]. This has changed the current focus of research to assessing the strength and frequency of convective downbursts and the ways these affect buildings differently from ABL gusts.

The current source of weather observations in the USA is the Automated Surface Observation System (ASOS), with almost 1000 stations installed between 2000 and 2007. These stations are operated, mainly at airports, by the National Weather Service (NWS), in collaboration with the Federal Aviation Authority and the Department of Defense, for whom the main priorities are for weather forecasting, aviation safety, and military activities, respectively. These priorities control how and what observations are made, reported, and archived. Other stakeholders, such as the construction and energy industries, have

no influence on this process and must find ways to make best use of the data that are collected. To be clear, this is not a criticism of the USA, since the same applies in all World Meteorological Organization (WMO) member countries. The ASOS system is currently unique in reporting observations continuously at one-minute intervals, enabling convective downbursts to be assessed.

Probably, the most important role of ASOS is to provide aircraft with weather information on demand. This requires the observations to be validated by a real-time quality control (QC) algorithm at the source station before onward transmission and archiving. These real-time data are also the source for the routine METAR weather reports and the derived daily, monthly, etc., summaries archived on the National Centers for Environmental Information (NCEI) servers. The previous studies [2,3] highlighted serious concerns about excessive false positive alerts by QC Test 10: the "low-speed non-meteorological event" detection algorithm [4] that leads to loss of valid data in mesoscale convective events. This test aims to remove spurious gust speeds caused by birds perching on the ASOS sonic anemometers [5,6].

The present study represents a fourth step by the author towards a comprehensive analysis of extreme wind speeds across CONUS caused by mesoscale convective events. The previous three steps in this quest were:

- Locating the exact positions of the ASOS anemometers [7] and assessing their local shelter in terms of the WMO exposure categories [8];
- Curating the archive of ASOS observations to detect non-meteorological artefacts, then correcting or removing them from the record [2];
- Classifying mesoscale convective events by type, corresponding to their presumed meteorological cause [3].

The purpose of this fourth step is to assess the extent to which the loss of valid data culled by the QC algorithms affects the extreme-value analysis of mesoscale convective gusts and what mitigation may be appropriate to limit underestimation of design gust speeds.

## 2. Materials and Methods

### 2.1. The ASOS One-Minute Interval Weather Observations

The ASOS network provides weather observations at 1-min intervals at 860 stations distributed across CONUS which are archived in the TD6405 and TD6406 databases at the NCEI. This is a most valuable resource because it permits study of many more mesoscale events over much longer observational periods than is possible with targeted measurement campaigns. Initially, from 2000, ASOS stations used Belfort cup and vane anemometers with a nominal 5 s gust response, most at 10 m above ground, but some at 26 feet (7.9 m). Between 2005 and 2010, they were gradually replaced by Vaisala sonic anemometers, averaged to give the WMO standard 3 s-mean gust, $\overline{V}_{3s}$. Having no moving parts, and heated in the winter to prevent icing, the sonic anemometers provide ideal perches for birds [5,6], which block the acoustic path, causing data loss, and which generate spurious large "spikes" of gust speed on landing and take-off.

Previous studies found the precise location and WMO exposure category of each anemometer mast [7] and curated the observations [2] to correct or remove common acquisition, transmission, and archiving errors [9]. Stations in WMO exposure category 1 require no corrections for local shelter; category 2 stations can be corrected by standard methods (e.g., [8], in which the shelter effect of each obstacle is modelled by a turbulent Gaussian wake); category 3 stations require calibration by measurement (e.g., wind tunnel tests); category 4 stations are sufficiently close to an obstruction to experience flow reversal. which is not correctable for that wind direction; while category 5 stations are so poorly exposed as to be uncorrectable. In the previous classification [3], 450 stations, comprising all WMO category 1 and enough category 2 stations to provide an adequate distribution across CONUS, were selected, and their locations are shown in Figure 1. The category 2

stations have not been corrected for local shelter as this does not affect their QC culling, which is the focus of this study.

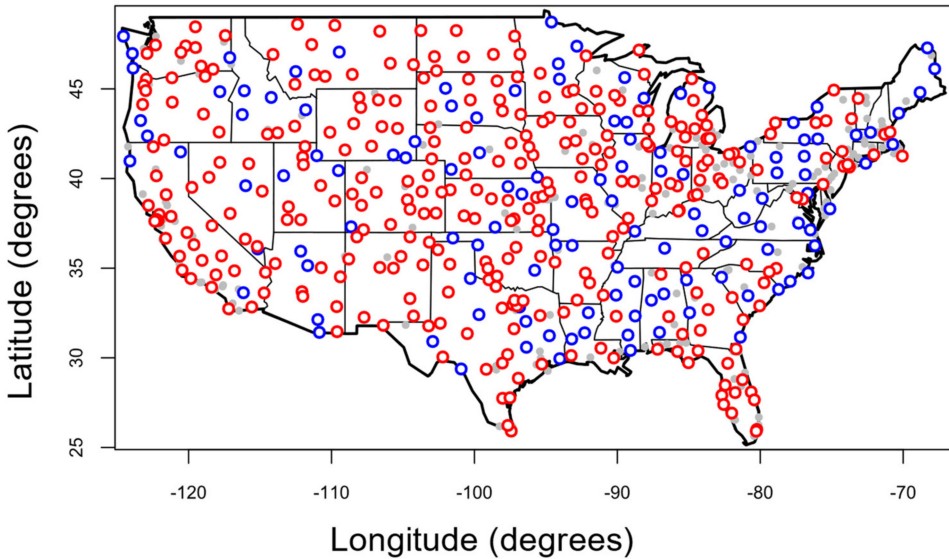

**Figure 1.** Locations of the 450 stations across CONUS. WMO category 1 stations are shown in red circles and category 2 stations are denoted by blue circles. Unused category 2 stations are denoted by grey dots.

In [3], all valid gust events >20 kn between 2000 and 2021 at these stations were sorted into six characteristic classes of valid events, each assigned to a presumed causal mechanism.

1.  Synoptic—comprising non-convective gusts generated by the ABL in synoptic-scale weather systems and near-neutral atmospheric stability. Steady, strong, locally stationary wind with low gust factor; little variation in direction and temperature; linear or no trend in atmospheric pressure; precipitation absent or continuous.
2.  Storm-burst—comprising short-duration, non-stationary, high gust-factor events in otherwise strong steady winds, sometimes with discernible variation in temperature or pressure. May include downdrafts from deep convection which penetrate through the gust structure of the ABL.
3.  Front-down—comprising convective downdrafts in the rear flank of active fronts, where the mean wind speed is decreasing from a higher steady value. Mean gust speeds after the peak are lower than before; direction veers or backs; temperature drops rapidly through the event; pressure increases temporarily when the downburst is directly over the station; otherwise, the variation is that expected for the passage of a front.
4.  Front-up—comprising convective downdrafts in the forward flank of active fronts, where the mean wind speed is increasing to a higher steady value. Mean gust speeds after the peak are lower than before; usually, but not always, associated with a sustained change in mean wind direction; otherwise, like Front-down.
5.  Thunderstorm—comprising downbursts from isolated thunderstorms, often in relatively light winds, where the initial wind speed and direction are restored after the event. Temporary sharp rise in gust speed over several minutes; temporary change in gust direction; sudden large drop in temperature; temporary increase in pressure when the downburst is directly over the station; otherwise, slight rise due to temperature drop; often a sudden burst of heavy rain.

*2.2. The ASOS Real-Time Quality Control*

The ASOS system reports mean and gust wind speeds in integer values of knots (1 kn = 0.5144 ms$^{-1}$), and it is essential for understanding how the QC works, as well as

to avoid unit bias to retain these original units. At the start of 2014, NWS added a new QC test algorithm—Test 10—to identify high gust speed events of non-meteorological origin occurring at low mean wind speeds. This was specifically to address the issue of bird-generated gusts, but it also finds numerous "spike" artefacts caused by acquisition or transmission glitches. Test 10 works by monitoring the running 2-min mean speed, $\overline{V}_{2\text{min}}$, and running 3-s mean speed, $\overline{V}_{3\text{s}}$, at 5 s intervals and raises an alert when:

$$\overline{V}_{2\text{min}} \leq 6 \text{ kn \& } \overline{V}_{3\text{s}} > 6 \text{ kn \& } \overline{V}_{3\text{s}} > 2.5 \times \overline{V}_{2\text{min}}, \tag{1}$$

after which, wind speed and direction observations are suppressed for 5 min. NWS reported [4] a 97% drop in erroneous reports at the cost of 0.033% loss of valid data. In late 2017, the $\overline{V}_{3\text{s}}$ threshold in (1) was raised from >6 kn to >13 kn to reduce the rate of false positives. Although the lost valid data comprise a very small fraction of the whole, its action in suppressing sudden gusts is strongly biased towards convective downbursts.

As justification for the present study, Figure 2 presents two examples of the ASOS 1-min interval timeseries through the two strongest recorded thunderstorm downbursts at Jackson International Airport, MS, both occurring while Test 10 applied at the 13 kn $\overline{V}_{3\text{s}}$ threshold. (Note that the chart sub-headings indicate ranks 19 and 23, respectively, because ranks 1 to 18 and 20 to 22 are all non-meteorological artefacts.)

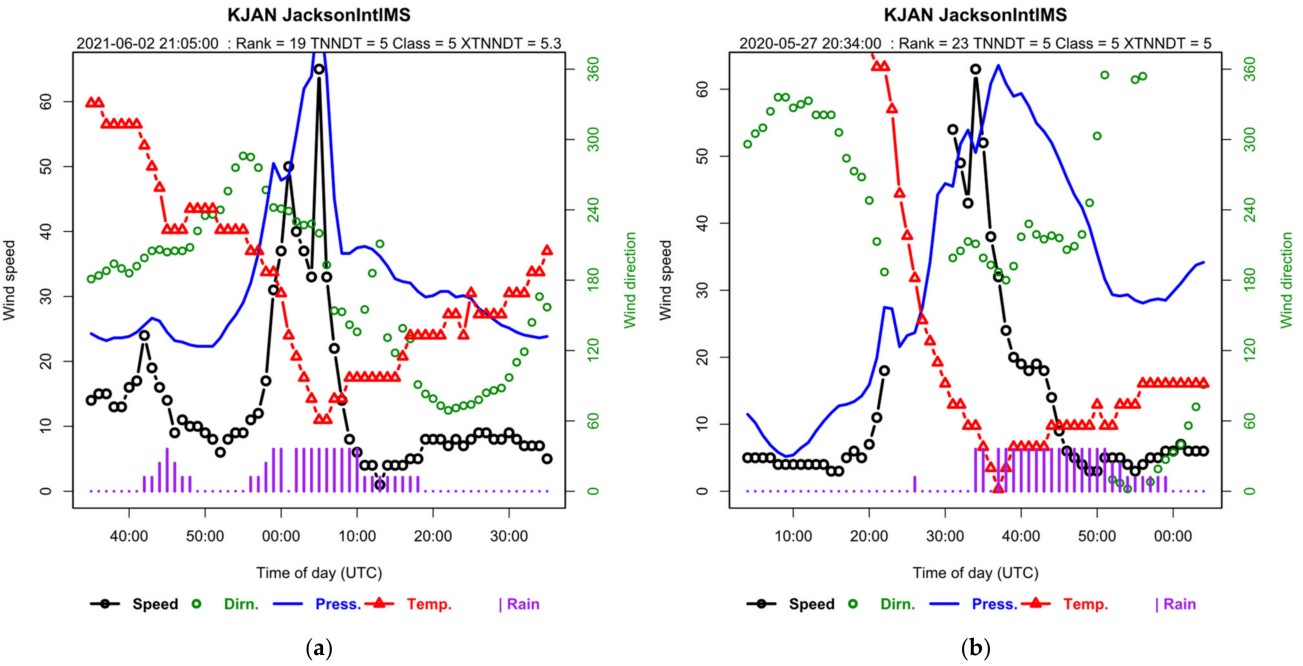

**Figure 2.** Extreme Class 5 Thunderstorm gust events at Jackson International AP, MS: (**a**) highest valid Thunderstorm gust; (**b**) second-highest valid Thunderstorm gust. The peak gust speed, $\hat{V}_{3\text{s}}$, in the previous minute (in knots) is shown by the (black) linked **o** symbols relative to the left-hand scale, and the 2-min mean direction (in degrees) by the (green) unlinked **o** symbols relative to the right-hand scale. Shown also are the contemporaneous atmospheric pressure (scale $\pm$ 0.05 in Hg) and temperature (scale $\pm$ 10 °F) anomalies and the amount of rain by the purple bars (none/light/medium/heavy). Time of day scale in minutes and seconds.

In Figure 2, the speeds are the peak 3 s-mean gust speed in the previous minute, $\hat{V}_{3\text{s}}$. No $\hat{V}_{3\text{s}}$ values are lost in (a) because $\overline{V}_{2\text{min}} > 6$ kn prior to the downburst, but $\hat{V}_{3\text{s}}$ values are culled from (b) for 5 min after $\overline{V}_{3\text{s}} > 13$ kn because $\overline{V}_{2\text{min}} < 6$ kn. Note that the maximum $\hat{V}_{3\text{s}}$ values of both events are concurrent with the minimum temperature and maximum pressure anomaly, so it is likely that the maximum $\hat{V}_{3\text{s}}$ value in (b) may be the true maximum for this event. However, in the more general case, where the true maximum

is excised by the cull, its value is unknown, and this is the main concern in assessing the impact of Test 10 false positives.

The pioneering insight of Gomes and Vickery [10] established the need to sort extreme events in mixed climates into classes by causal mechanism for separate analysis. Of the 5 event classes defined for the USA in [3] and listed earlier, only Class 4: Front-up events, which correspond to the arrival of an active cold front, and Class 5: Thunderstorm events in otherwise light winds are potentially sensitive to Test 10 by virtue of low initial $\overline{V}_{2min}$ values. Figure 3 presents the ensemble-averaged gust speed, temperature, and pressure timeseries of these two classes. For both classes, there is a rapid fall in temperature and rise in pressure concurrent with the peak gust. The gust speed timeseries differ in that the speed quickly recovers to the initial value for Class 5 but remains high much longer for Class 4. The individual gust events, e.g., Figure 2, are much more variable than these ensemble-averages suggest. Those recovering slower are clearly Class 4, those recovering faster are clearly Class 5, but there is some uncertainty in the Class of intermediate events. The overall classification accuracy of the neural network in [3], which interprets the gust speed, temperature, and pressure timeseries, is claimed to be better than 95%.

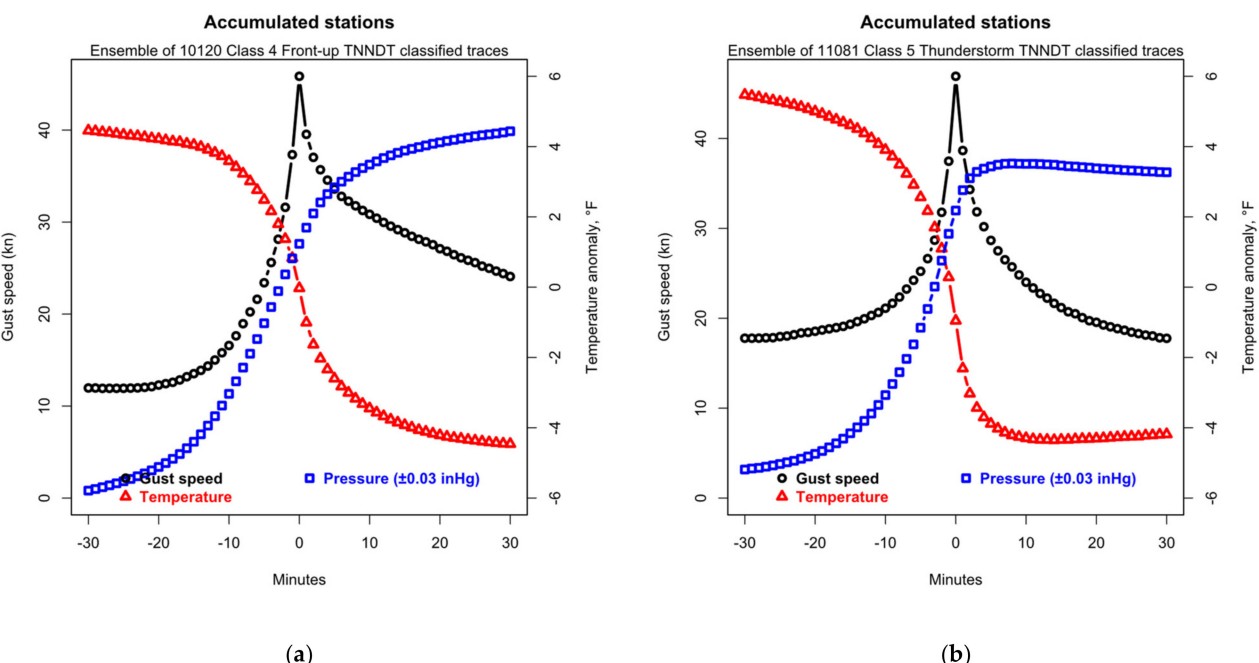

(**a**)    (**b**)

**Figure 3.** Ensemble-averaged gust speed $\hat{V}_{3s}$, temperature. and pressure timeseries for all Class 4 and 5 gust events at the 450 stations in Figure 1: (**a**) Class 4; (**b**) Class 5.

On average, the initial $\overline{V}_{2min} > 6$ kn for both classes, so only subsets of these events with low initial speeds will be affected by Test 10—potentially, 0.76% of Class 4 and 0.65% of Class 5 at the 6 kn $\overline{V}_{3s}$ threshold; and 0.23% of Class 4 and 0.33% of Class 5 at the 13 kn threshold. These percentages are small but are an order of magnitude greater than the 0.033% overall loss of valid data [4], highlighting the bias in Test 10 false positives towards convective gust events.

### 2.3. The XIMIS Method of Extreme-Value Analysis

The XIMIS method [11], fitted by weighted least-mean-squares (wLMS) and displayed on Gumbel axes, was preferred [12] over other extreme-value analysis (EVA) methods for the following reasons:

1.    Probability estimated from the order statistics are for independent events following a Poisson recurrence process model leading to a Fisher–Tippett Type 1 (FT1) asymptote;

2. The test for validity of the Poisson model is exponentially distributed inter-arrival times [13], which is simple to implement;
3. The order statistics are ranked downwards from the highest value, so XIMIS is the only method (apart from ACER [14]) that works for left-censored data where the population, $N$, is unknown;
4. Classified gust event data from [3] are left-censored at $\hat{V}_{3s} > 20$ kn;
5. Asymptotic convergence is not a requirement, i.e., XIMIS is a penultimate method;
6. Confidence limit outliers are easily detected, and their contribution removed.

The XIMIS extreme model can use any fitting method, but wLMS was used here with the recommended fitting weights [11] to account for statistical variance.

*2.4. XIMIS Methodology*

The variate, $V$, is normalized by the Gumbel "reduced variate" [15], $y_V$:

$$y_V = (V - U)/b, \tag{2}$$

where $U$ is the mode and $b$ is the dispersion. The annual probability of exceedance, $\Phi_m$, for the $m$-th rank from the top is linearized in the FT1 model by the Gumbel "plotting position" [15], $y_m$:

$$y_m = -\ln(-\ln(\Phi_m)), \tag{3}$$

so that a plot of $y_V$ as abscissa against $y_m$ as ordinate presents the FT1 asymptote as a straight line of slope 1 through the origin. In practice, $V$ is usually plotted as abscissa, whereupon the slope is $b$ and the intercept is $U$. Deviation from the asymptote due to non-convergence resolves as a curve which may be fitted by any method, but, when necessary, XIMIS seeks to linearize this by the transformation:

$$y_V = (V^w - U^w)/b^w, \tag{4}$$

which is exact when the upper tail of the parent distribution of $V$ has Weibull equivalence with shape parameter $w$.

XIMIS uses the mean plotting position, $\bar{y}$, given in [15], Eq. 4.2.1 (11), for each $m$ by the recursive formula:

$$\bar{y}_{m+1} = \bar{y}_m - 1/m \text{ with } \bar{y}_1 = \gamma + \ln(R), \tag{5}$$

where $\gamma = 0.5772$ (Euler's constant) and $R$ is the observation period in years. Equation (5) is exact for an Exponential parent ($w = 1$) but, when $w \neq 1$, progressively accumulates error at large $m$, which is avoided by left censoring. The corresponding variances, $\sigma_m^2(y)$, are given in a similar manner [15], Eq. 4.2.1 (13), by

$$\sigma_{m+1}^2(y) = \sigma_m^2(y) - 1/m^2 \text{ with } \sigma_1^2 = \pi^2/6. \tag{6}$$

The accuracy of (5) and (6) is assessed in [12].

The distributions of plotting position around the mean, $\bar{y}_m$, are also the distributions of the $m$-th highest values for repeated samples of the observation period, $R$. In practice, as there is only one observation period, the distributions of the $m$-th highest values are used to assess the confidence that can be placed in the individual values of that one sample. The PDF of the $m$-th highest values is given in [15], Eq. 5.3.2 (2), by

$$p_m(y_m) = \frac{m^m}{(m-1)!} \exp\left(-my_m - me^{-y_m}\right), \tag{7}$$

and the corresponding CDF of the $m$-th highest values is given in [15], Eq. 5.3.2 (3), by

$$P_m(y_m) = P_1{}^m(y_m) \sum_{v=0}^{m-1} \frac{m^v e^{-vy_m}}{v!}. \tag{8}$$

Note that $p_m(y)$ and $P_m(y)$ are continuous functions for each $m$-th extreme but are evaluated by (7) and (8) only for the discrete ranks. Confidence limits for an XIMIS model are evaluated by solving (8) for $y_m$ at each desired confidence level $P_m$ and for the median, $\breve{y}_m = y_m(P_m = 0.5)$, then applying their differences, $\delta y_m(P_m) = y_m(P_m) - \breve{y}_m$, to either side of the model. Owing to the large powers of $m$ encountered when solving (8), the practical range using 64-bit floating point arithmetic is limited to $m \leq 100$.

### 2.5. XIMIS Charts

All the XIMIS charts which follow are presented on the conventional Gumbel axes [15], with maximum gust speed $\hat{V}_{3s}$ as abscissa and $y = -\ln(-\ln(\Phi))$ as ordinate. When all the integer $\hat{V}_{3s}$ values are plotted, the many tied values create a staircase effect with steps wider than the confidence limits. For clarity of presentation, only one point was plotted for each integer $\hat{V}_{3s}$ at the ensemble mean plotting position of each set of ties, $\langle \overline{y}_m \rangle_{\hat{V}}$. All observations above the lowest indicated speed were included in the FT1 model fit, shown by the straight lines, which was fitted by wLMS using $1/\sigma_m^2$ as the fitting weights [11]. Confidence limits of 5% and 95% on $\hat{V}_{3s}$ are shown by the chained curves. Strictly, the confidence is in the value of $y_m$ assigned to each rank [11], and so indicates the uncertainty in the abscissa. However, following the convention of Gumbel [15], here they are applied through (2) to indicate the uncertainty in the ordinate $\hat{V}_{3s}$ at the mean plotting positions. Note that these limits indicate the confidence of each individual observation, i.e., whether an observation is an outlier that should be discounted [15]. All the observations contribute to the fitted model, inversely weighted by their variance, so the model confidence limits are much tighter. Model confidence can be evaluated numerically by the "jackknife" or by the bootstrap [16], but this is outside the scope of the present study and is addressed elsewhere [12].

### 2.6. CONUS Mapping

All the maps of values over CONUS which follow are presented as a graded color scale on longitude–latitude axes. This is overlaid by the CONUS and State boundaries and the locations of the individual stations. Strictly, the values apply only at the station locations, but barycentric linear interpolation has been used to fill in the spaces between. Owing to this interpolation, the effect of a single station with an unusually high or low value spreads out towards adjacent stations. Wind speed maps in codes and standards, as well as design wind speeds for a specific location, are typically interpolated between adjacent stations, so this is compatible with this convention.

## 3. Results
### 3.1. Observed Effect of Test 10 from 2014–2021

Observations from 2014 to late 2017 were subject to Test 10 at the 6 kn $\overline{V}_{3s}$ threshold, and those from 2018 onwards to the 13 kn $\overline{V}_{3s}$ threshold. A Test 10 false positive culls 5 min of valid wind speed and direction data, sometimes longer if $\overline{V}_{2min}$ falls below its trigger threshold again. This should not affect the timeseries of temperature and pressure. Class 4 and 5 gust events over the whole period 2000–2021 were searched for gaps in speed and direction greater than 4 min. Those gaps immediately preceded by $\overline{V}_{2min} \leq 6$ kn and where temperature and pressure remained continuous were counted as a Test 10 false positive, while those >6 kn were counted as due to other causes. The average annual rates of gaps per station for Class 4 and 5 events are plotted in Figure 4. This shows a sharp rise in the rate of gaps on the introduction of Test 10 which continues rising slowly thereafter, while the rate due to other causes remains reasonably constant. Some may have been caused by Test 10 after $\overline{V}_{2min}$ fell below the 6 kn threshold briefly in the 1-min overlap between observations.

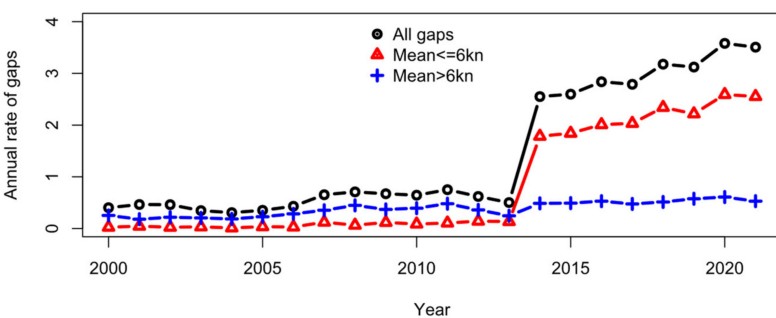

**Figure 4.** Average annual rate of data gaps found in Class 4 and 5 events, 2000–2021.

The annual rates of Test 10 culls of valid Class 4 and 5 events are mapped across CONUS in Figure 5 for 2014–2017 and for 2018–2021, representing each threshold. The highest rates occur in the high-altitude stations of UT, CO, AZ, and NM, with the maximum at KASE Aspen, CO. The annual rates of data gaps from other causes shown in Figure 6 are an order of magnitude lower and more randomly distributed.

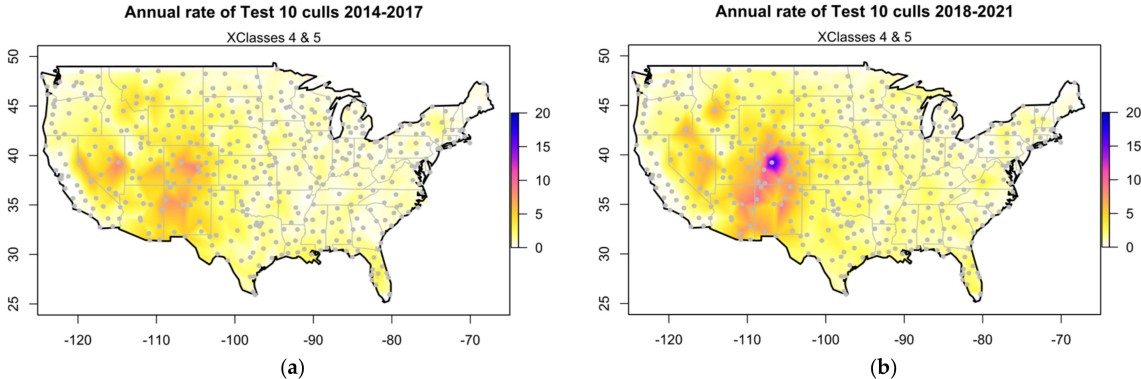

**Figure 5.** Observed annual rates of Class 4 and 5 events culled by Test 10: (**a**) 2014 to 2017; (**b**) 2018 to 2021.

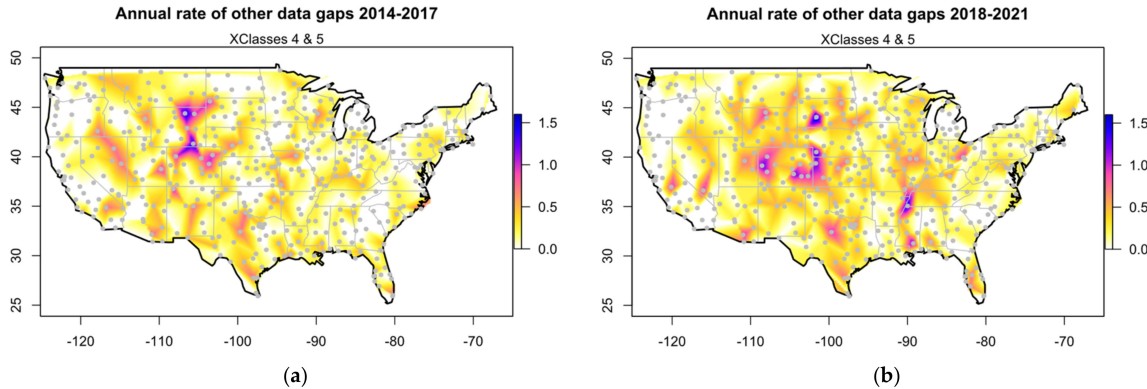

**Figure 6.** Actual annual rates of gaps from other causes in Class 4 and 5 events: (**a**) 2014 to 2017; (**b**) 2018 to 2021.

### 3.2. Simulating Test 10 Using Data before 2014

The major issue with the Test 10 false positives after 2013 is that the culled observations are unrecoverable, so it is uncertain whether the peak gust is included and, if it is, how much it is underestimated by the surviving observations. The gust events which occurred

before the introduction of Test 10 are suitable for retrospectively assessing its effect at both 6 kn and 13 kn thresholds. The only significant difference is that the reported observations of $\hat{V}_{3s}$ are at 1-min intervals, whereas Test 10 operates on $\overline{V}_{3s}$ at 5 s intervals. This has two potential counteracting consequences:

1. The 5 s intervals give 12 opportunities in each minute for the running $\overline{V}_{2min}$ to fall to 6 kn or less and trigger Test 10, whereas the observations give only one which should lead to fewer simulated culls;

2. The values of $\hat{V}_{3s}$ in the gust events are the largest observed in the previous 1 min, not $\overline{V}_{3s}$ in the previous 5 s, so should lead to more simulated culls.

The simulated annual rates of Class 4 and 5 false positives at either Test 10 threshold are mapped across CONUS in Figure 7. At the 6 kn $\overline{V}_{3s}$ threshold (left), the highest rates again occur in the high-altitude stations of UT, CO, AZ, and NM, with the maximum at KASE Aspen, CO. At the 13 kn $\overline{V}_{3s}$ threshold (right), the rates are considerably reduced. While the geographical distributions are the same as observed in Figure 5, the trend in rate between thresholds is reversed.

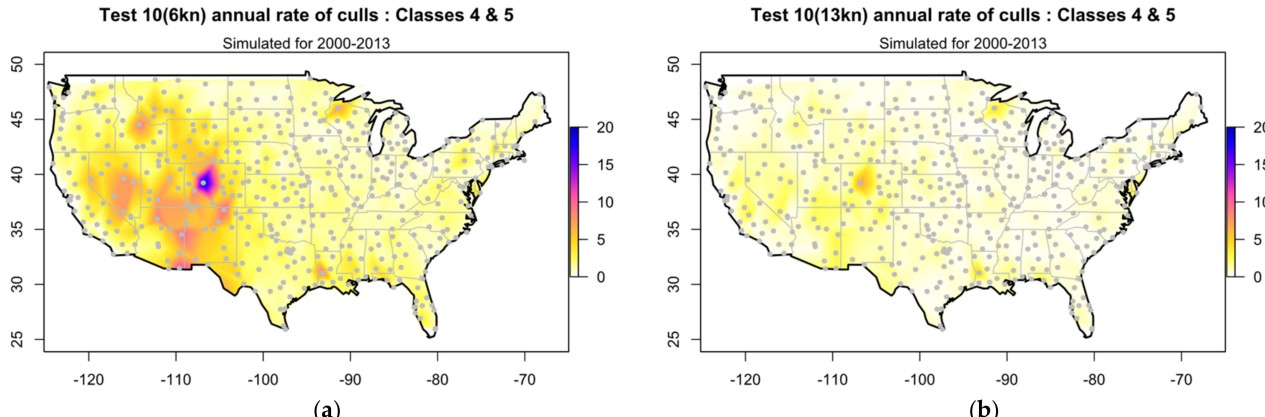

**Figure 7.** Simulated annual rates of Class 4 and 5 Test 10 false positives from 2000–2013: (**a**) 6 kn threshold; (**b**) 13 kn threshold.

The period culled by Test 10 does not always include the peak gust. The distribution of the simulated annual rate of Class 4 and 5 $\hat{V}_{3s}$ values that are underestimated by culling the peak is shown in Figure 8 to have a similar geographical pattern as Figure 5, but only 10% of the culls result in an underestimate of the peak gust. The simulated rate is lower for the 13 kn $\overline{V}_{3s}$ threshold, which was the intent of the threshold change. The underestimated values of $\hat{V}_{3s}$ caused by the ~10% of simulated Test 10 false positives that cull the peak are shown in the Q–Q plots of Figure 9 for both Classes and thresholds. As these are in integer knots, each point may represent multiple events. The solid line shows the linear regression, and the chained line denotes 1:1 correspondence. The scatter around the regression line is too large and random to allow any reasonable correction. Half the points lie above the regression line and would produce safe overestimates. However, half lie below and would still underestimate some outliers by up to ~20 kn. It is therefore necessary to directly assess how these false-positive culls would affect the distribution of extremes.

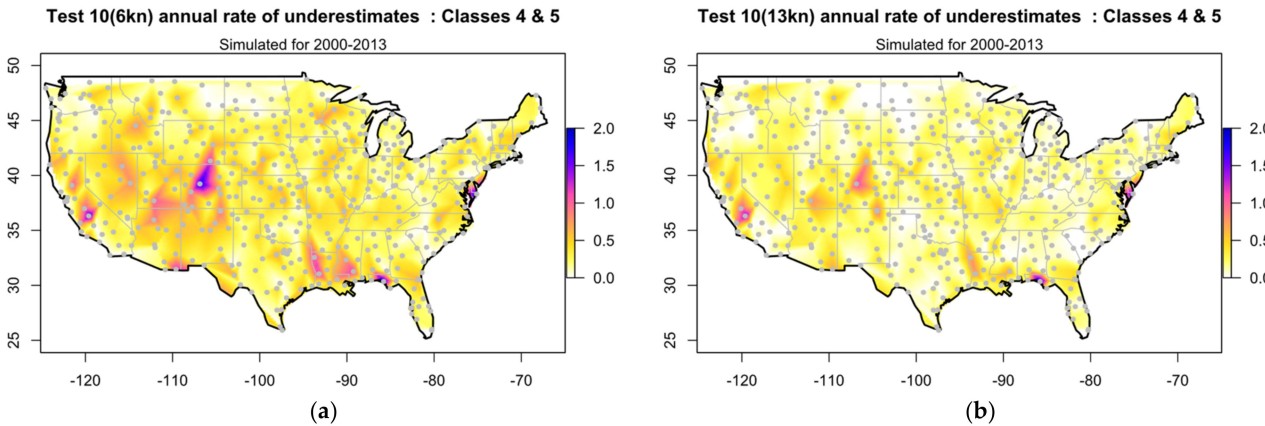

**Figure 8.** Simulated annual rates of Class 4 and 5 $\hat{V}_{3s}$ underestimated by Test 10 from 2000–2013: (**a**) 6 kn threshold; (**b**) 13 kn threshold.

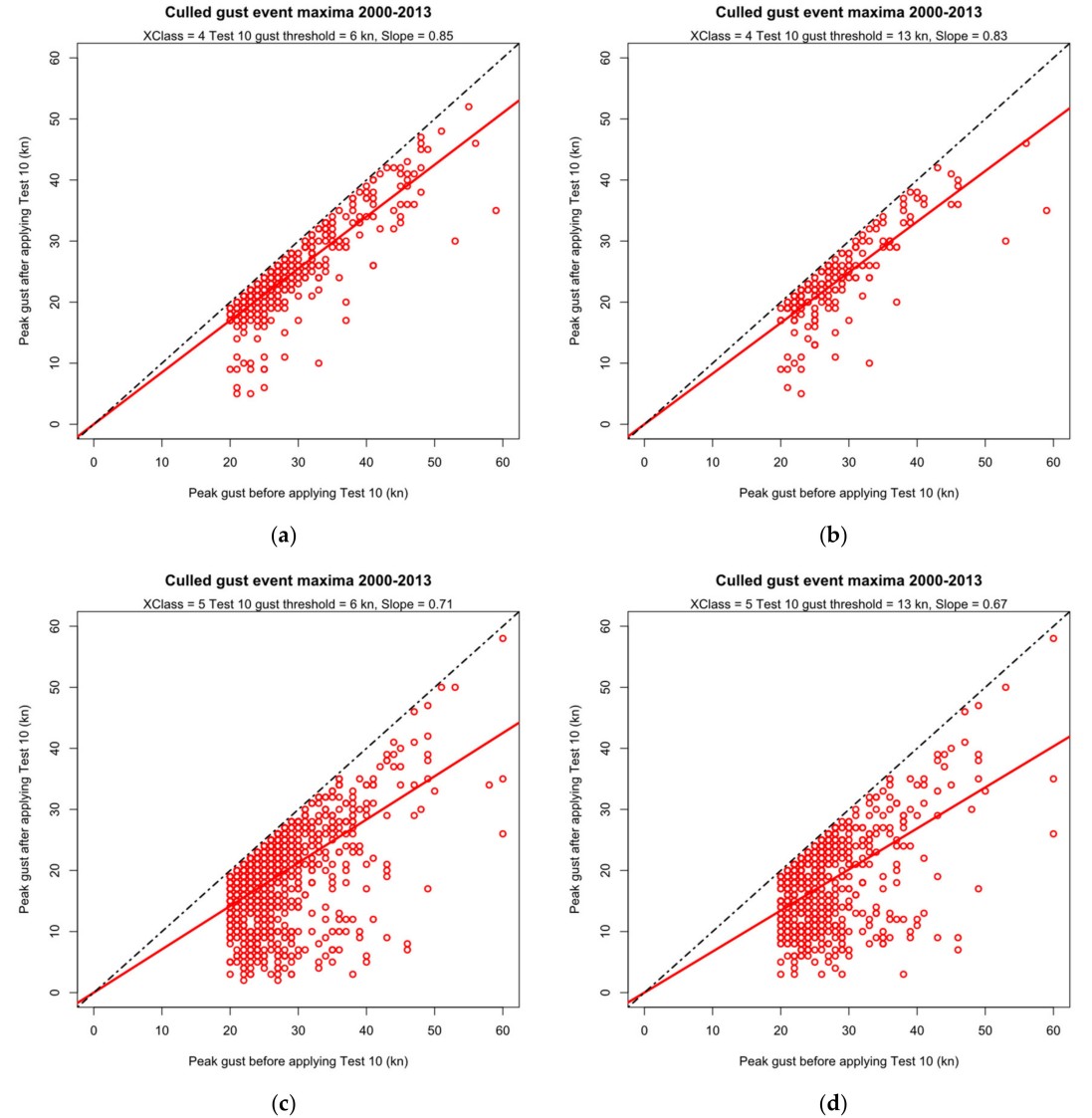

**Figure 9.** Actual $\hat{V}_{3s}$ and simulated Test 10 underestimates for Class 4 and 5 from 2000–2013: (**a**) Class 4, 6 kn threshold; (**b**) Class 4, 13 kn threshold; (**c**) Class 5, 6 kn threshold; (**d**) Class 5, 13 kn threshold. Dashed black line indicates 1:1 correspondence. Solid red line indicates linear regression.

### 3.3. Simulating the Effect of Test 10 on XIMIS Extremes 2000–2013

#### 3.3.1. CONUS Superstation

The ensemble of all independent events at a group of stations representative of an area into a "superstation" was first used by Peterka [17] to reduce the sampling error from short records of fastest-mile wind speeds across the US Midwest. Figure 10 presents the XIMIS charts for Class 4 and 5 events amalgamated from all 450 stations, corresponding to 8360 station-years of observations. The charts show the fit for the original observations and for both Test 10 thresholds. Both charts show a good linear fit to the FT1 asymptote, requiring no correction for non-convergence offered by (4). All observations lie within the confidence limits. If the Class characteristics were uniform across CONUS—which, of course, they are not—the highest observed $\hat{V}_{3s}$ in each Class could be interpreted as having an 8360-year mean recurrence interval (MRI). The validity of the superstation concept depends on two contradictory conditions: (1) that the individual stations are well separated to ensure independence; and (2) that the area they represent is sufficiently small for uniformity.

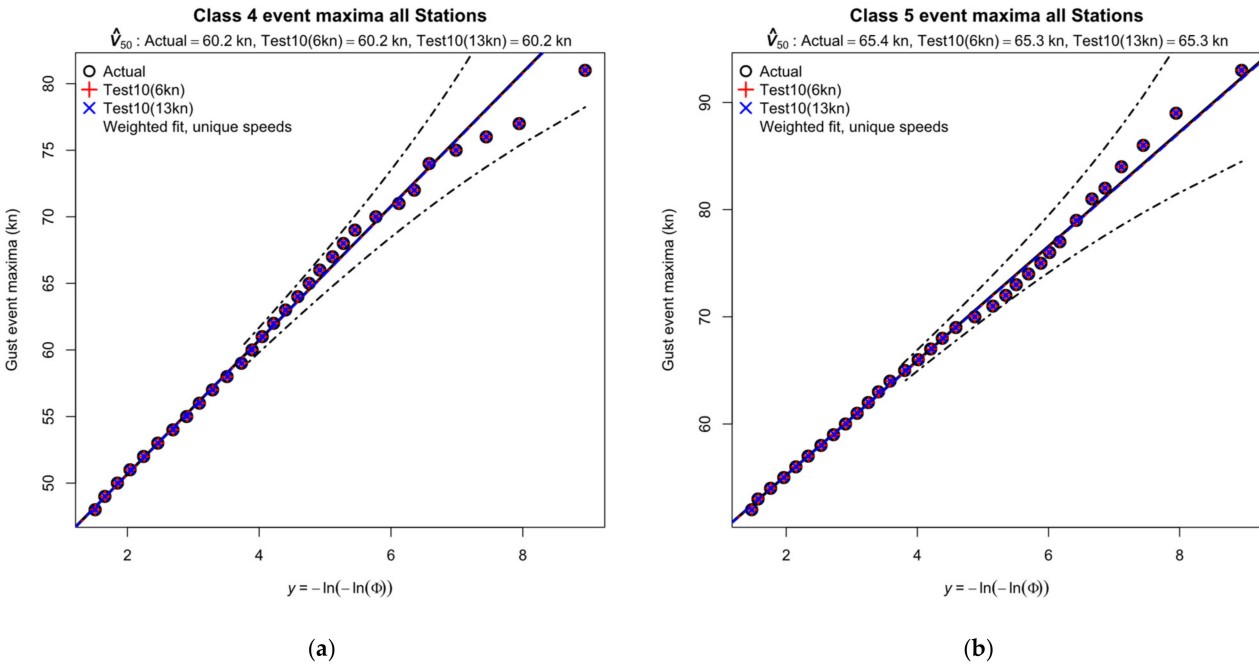

(**a**) (**b**)

**Figure 10.** XIMIS analysis of pre-2014 gust events for the CONUS superstation: (**a**) Class 4, Frontal downburst; (**b**) Class 5, Thunderstorm downburst.

The principal conclusions from Figure 10 are that the FT1 asymptote is a good model for these strongly convective gust events and that the impact of Test 10 on the 50-year MRI gust speed $\hat{V}_{50}$ is minimal for the CONUS superstation. The provenance of these conclusions is tracked for the highest observed $\hat{V}_{3s}$, via the corresponding US state superstations, to the individual ASOS stations in the following sections.

#### 3.3.2. The Kansas and Illinois State Superstations

The highest observed $\hat{V}_{3s}$ occurred in Kansas for Class 4 and in Illinois for Class 5. Figure 11 presents the corresponding XIMIS charts for KS (290 station-years) and IL (194 station-years) state superstations. Again, both charts show a good linear fit to the FT1 asymptote with all points within the confidence limits, and no discernible influence of Test 10 on $\hat{V}_{50}$. The highest observed $\hat{V}_{3s}$ now lie above the fitted asymptote and projection onto the fitted model gives an MRI of 667 years for Class 4 in KS and 686 years for Class 5 in IL, a large reduction from the CONUS values.

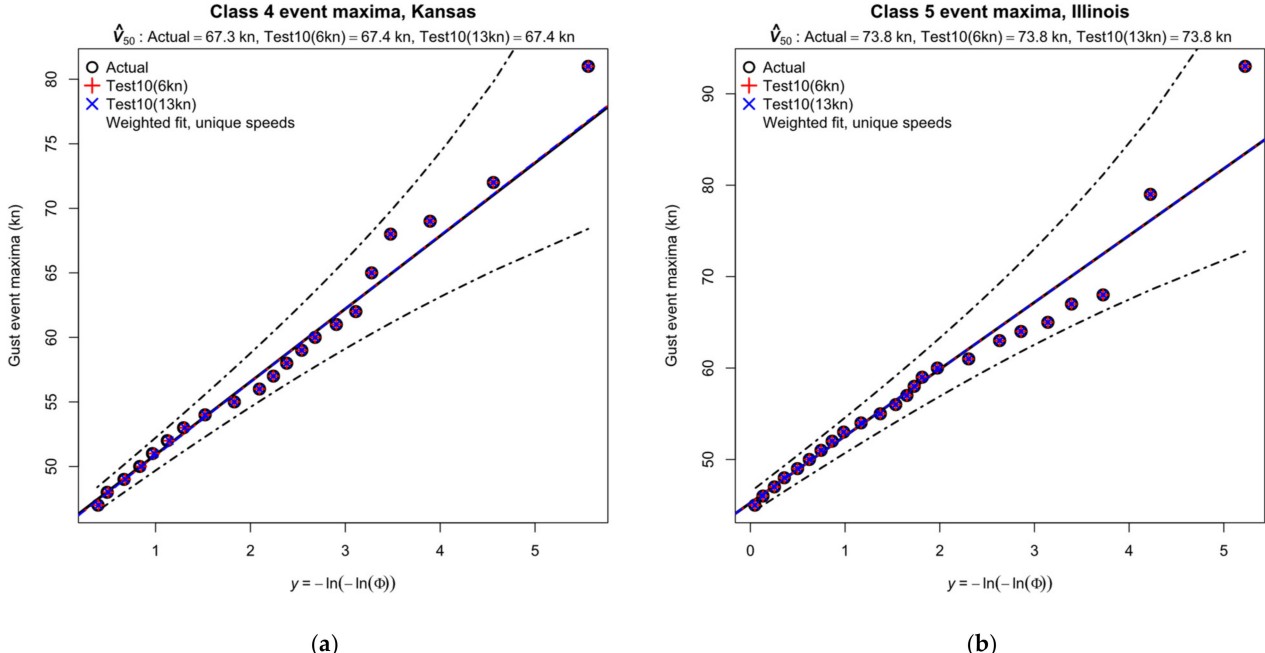

**Figure 11.** XIMIS analysis of pre-2014 gust events for Illinois superstations: (**a**) Class 4 for Kansas and (**b**) Class 5 gust events.

3.3.3. KGCK Garden City, KS, and KMLI Moline, IL

The highest observed $\hat{V}_{3s} = 82$ kn for Class 4 occurred at KGCK Garden City, KS, and $\hat{V}_{3s} = 93$ kn for Class 5 at KMLI Moline, IL. Figure 12 presents the XIMIS charts for these two stations. Allowing for the higher statistical variance due to the smaller populations, both charts still show a good linear fit to the FT1 asymptote with all points just within the confidence limits and no discernible influence of Test 10 on $\hat{V}_{50}$. The corresponding MRI of the highest observed $\hat{V}_{3s}$ are now 36 years and 213 years, respectively, with the latter only just within the confidence limits. Figure 13 presents the 1-min interval time series in the same format as Figure 2.

- The event at KGCK (left) shows the typical Class 4 characteristics of an active cold front:
  - The peak gust occurs at the forward flank of the front and recovers to a sustained value greater than the incident value;
  - A sudden sustained change in direction is contemporaneous with the peak gust;
  - The temperature falls through the event to a sustained minimum ~15 min after the peak gust, whereas in a typical Class 5 event, e.g., Figure 2, the temperature minimum typically coincides with the peak gust;
  - The pressure rises to a sustained value;
  - A burst of heavy rain starts at the peak gust and lasts for 10–20 min.
- The 93 kn gust at KMLI occurred during a severe outbreak of thunderstorms and tornados across MO, IL, and AR. It is reported by the NOAA Storm Prediction Center [18] wind report for the day as a non-tornadic event which destroyed the primary ASOS anemometer but was recorded by the back-up equipment. It differs from the typical Class 5 thunderstorm events in Figure 2 in some important respects:
  - The $\hat{V}_{3s}$ timeseries shape is typical of a Class 5 thunderstorm event but the duration is untypically short;
  - The transient change in direction is untypically short;
  - There is a transient rise in temperature instead of the expected fall;
  - The transient changes in pressure are untypically abrupt;
  - The continuous rainfall is interrupted at the time of the peak gust.

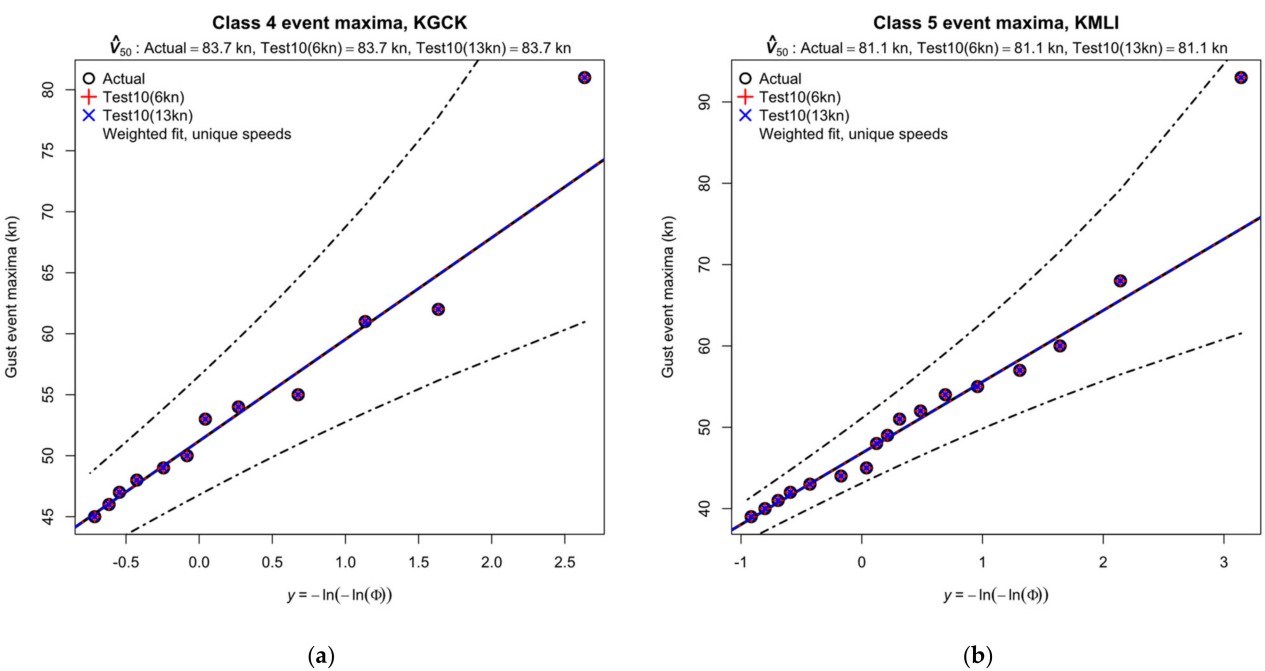

(**a**)       (**b**)

**Figure 12.** XIMIS analysis of pre-2014 gust events: (**a**) Class 4 at KGCK and (**b**) Class 5 at KMLI.

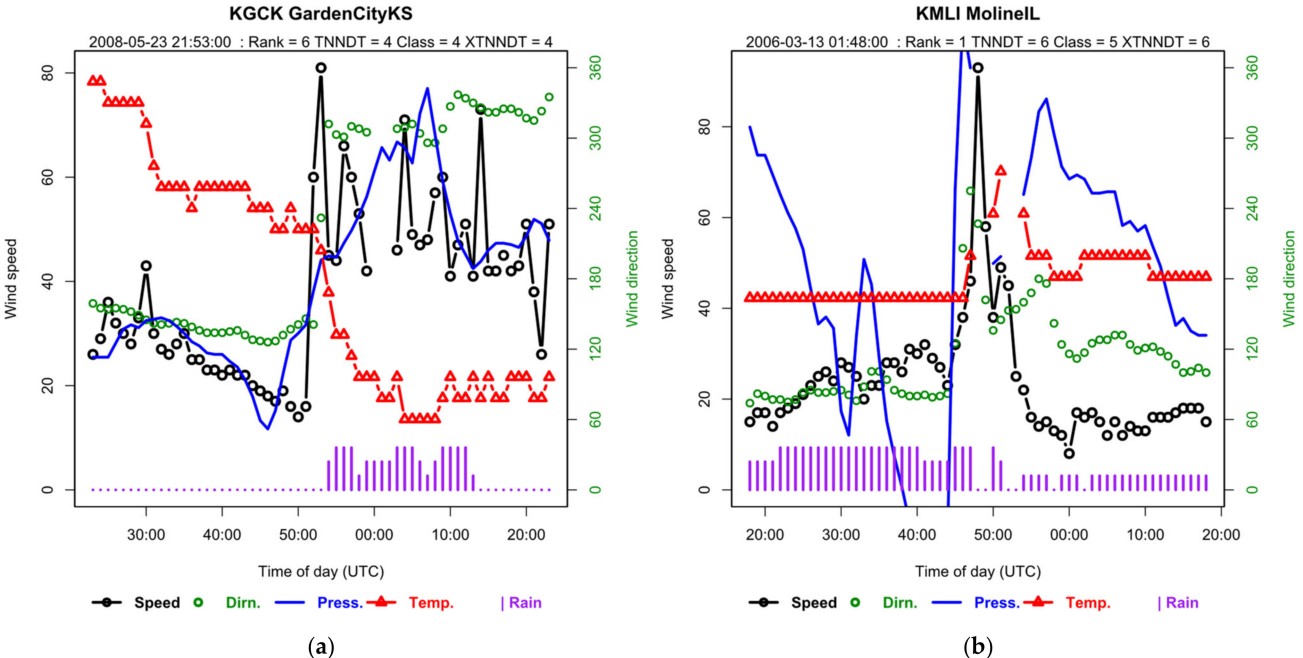

(**a**)       (**b**)

**Figure 13.** Gust event timeseries for highest observed $\hat{V}_{3s}$ gust: (**a**) Class 4 at Garden City, KS, and (**b**) Class 5 at Moline, IL. The peak gust speed, $\hat{V}_{3s}$, in kn is shown by the (black) linked **o** symbols relative to the left-hand scale and the 2-min mean direction in degrees by the (green) unlinked **o** symbols relative to the right-hand scale. Shown also are the contemporaneous atmospheric pressure (scale $\pm$ 0.05 in Hg) and temperature (scale $\pm$ 10 °F) anomalies, and the amount of rain is denoted by the purple bars (none/light/medium/heavy). The time of day scale is in minutes and seconds.

This is cited as an example of a "warm" event in [3], which includes local thermally induced events and represent 7.3% of Class 5. The characteristics of this event are sufficiently atypical of the training set for both neural networks in [3] for the event to have been classified as non-meteorological in origin (TNNDT = 6 and XTNNDT = 6 in the chart

subheadings). Reclassification to Class 5 was made by visual inspection of the timeseries and confirmation by SPC [18].

### 3.3.4. Influence of Test10 at All Stations

Although simulating Test 10 appears to have no discernible influence on most stations, analysis of all stations reveals that about 10% are affected. Figure 14 shows the distribution of error in the values of $\hat{V}_{50}$ predicted by XIMIS for both Classes and thresholds. These distributions are skewed towards underestimation, with the largest underestimate of $\hat{V}_{50}$ for Class 4 at KBKB Fort Polk Fullerton, LA, and for Class 5 at KTQE Tekamah, NE. Raising the threshold to 13 kn reduces the frequency of errors for both Classes and the range of errors for Class 4.

Underestimation is a concern for the design of safe structures. The XIMIS charts for the largest underestimate of each Class are presented in Figure 15.

- At KBKB, the underestimate of 4.3 kn (10.7%) for Class 4 at the 6 kn threshold is principally caused by culling the $m = 1$ event, which removes the peak value and demotes the event rank to $m = 22$. The effect is amplified by the small population of Class 4 events at this station and the large statistical variance denoted by the wide confidence limits. Raising the threshold to 13 kn restores the culled peak and $\hat{V}_{50}$ is then marginally (1%) overestimated. KBKB is a WMO category 2 station and some of this underestimate may be due to shelter by local obstacles.
- At KTQE, the underestimate of 5.2 kn (5.7%) for Class 5 at both thresholds is principally caused by culling the $m = 3$ and 11 events. Raising the threshold to 13 kn does not restore these missing events. KTQE is a WMO category 1 station, which rules out any effect of local shelter.

The distribution of $\hat{V}_{50}$ underestimates for both Classes and thresholds are mapped across CONUS in Figure 16. The locations of errors appear confined to individual randomly located stations, although the larger errors resolve as patches around each affected station due to the barycentric linear interpolation. There is no discernible correlation between the locations of Class 4 and Class 5 locations. The reduction in errors on raising the threshold from 6 kn to 13 kn is evident, more so for Class 4 than Class 5.

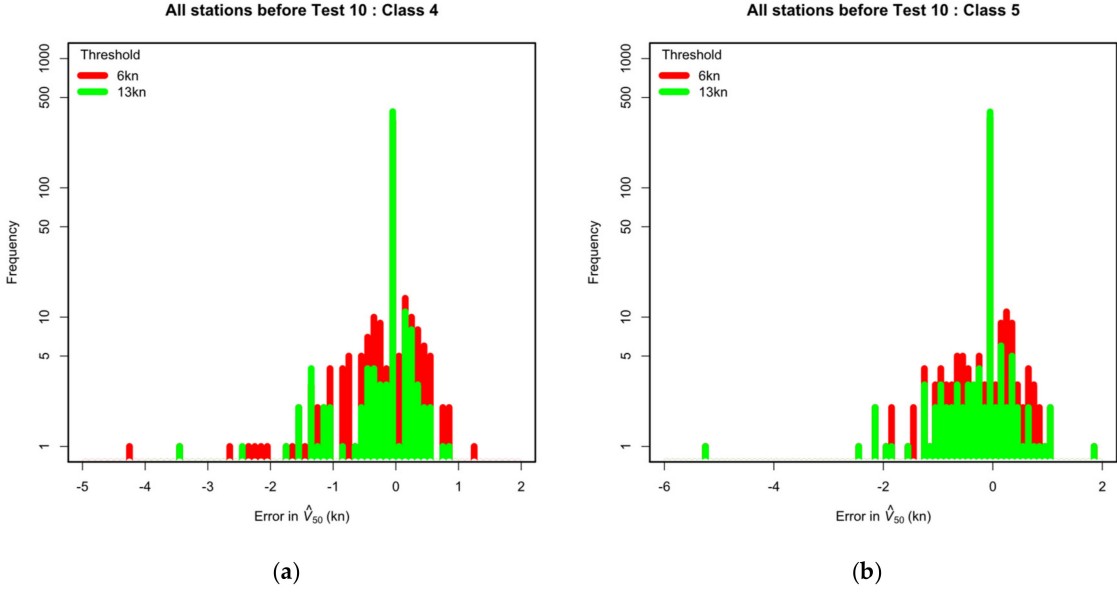

(**a**)  (**b**)

**Figure 14.** Effect of simulated Test 10 threshold value on error distribution of $\hat{V}_{50}$ for: (**a**) Class 4 and (**b**) Class 5.

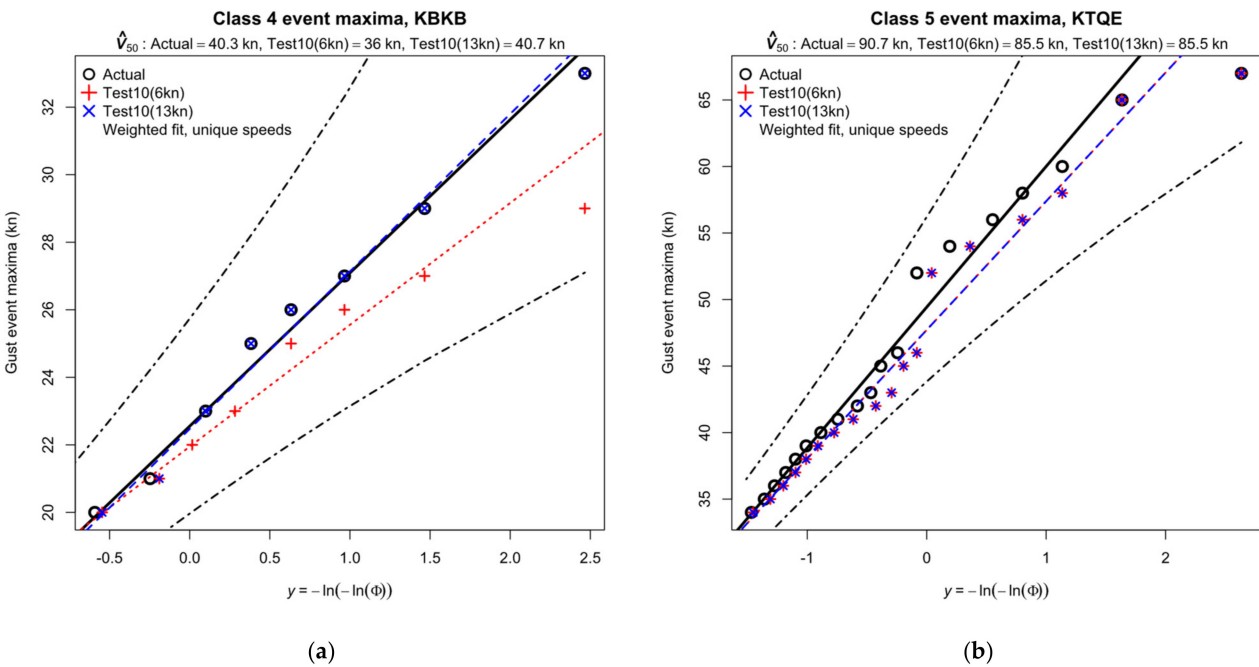

(**a**) (**b**)

**Figure 15.** XIMIS analysis of peak $\hat{V}_{3s}$ in pre-2014 gust events: (**a**) Class 4 at KBKB and (**b**) Class 5 at KTQE.

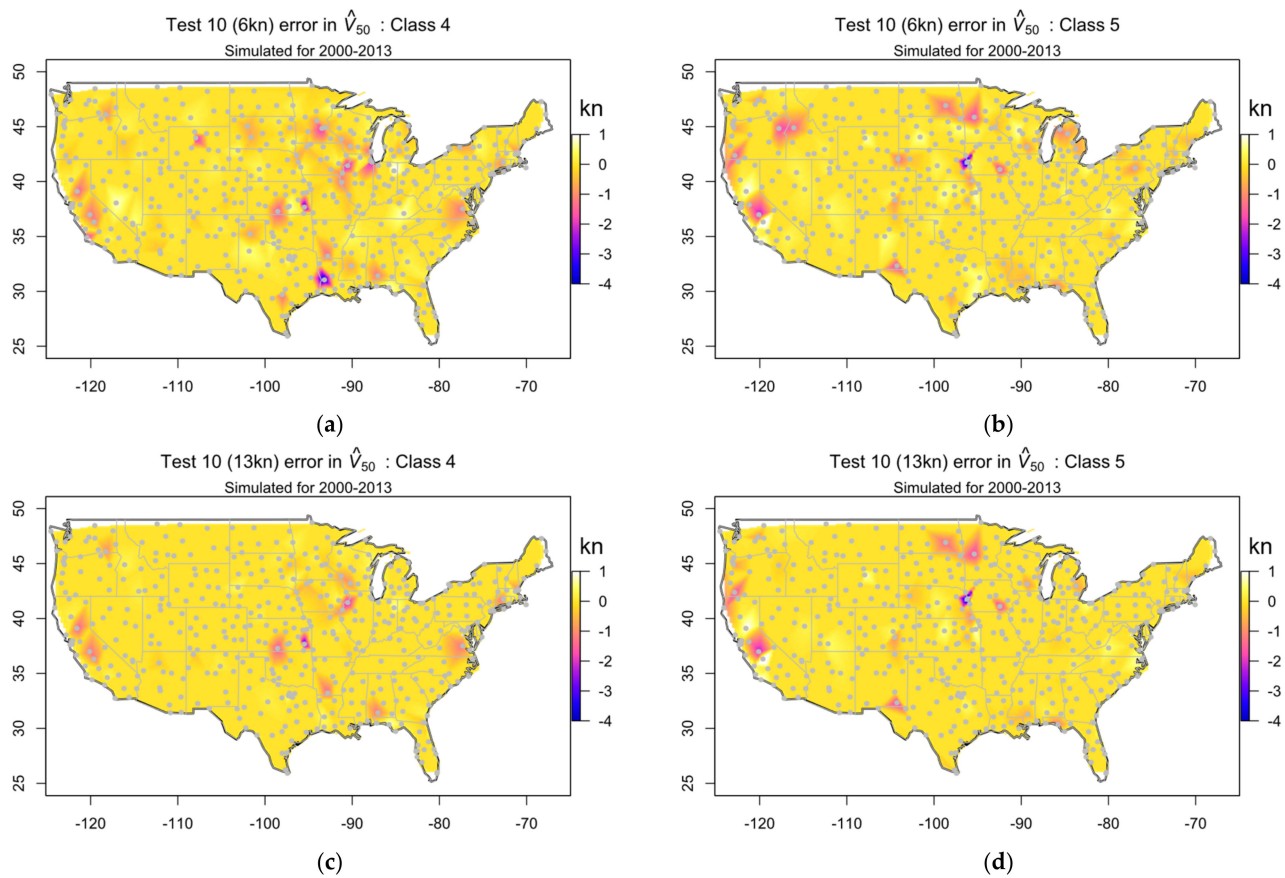

(**c**) (**d**)

**Figure 16.** Distribution of error of $\hat{V}_{50}$ (in kn) from Test 10 applied to pre-2014 gust events: (**a**) Class 4, 6 kn threshold; (**b**) Class 5, 6 kn threshold; (**c**) Class 4, 13 kn threshold; (**d**) Class 5, 13 kn threshold.

### 3.4. Predicted 50-Year Maximum Gust Speeds

The values of $\hat{V}_{50}$ predicted by XIMIS for all stations over the observation period 2000–2021 are mapped across CONUS in Figure 17. These are designated as "raw" in that no corrections have been applied to compensate for the effect of Test 10 false positives. Again, potential underestimates resolve as patches around single stations, although these do not correlate well with the locations in Figure 16, and some may be genuinely low values and others because WMO category 2 station exposures have not been corrected. Locations of underestimates that would have occurred in 2000–2013, had QC Test 10 operated then, do not predict where underestimates occur in 2014–2021. Neither do the locations where the observed rate of culls is high (Figure 5) illustrating the inherent uncertainty of the impact of Test 10. There is a complex balance between militating effects, such as the rate of all culls, and mitigating effects, such as the small proportion that result in underestimates and the lower weighting given to the smaller ranks (higher values) at each station. Stations with the highest values are least influenced by Test 10, implying that the highest peaks start from initial mean values higher than the threshold. The effect on risk of exceedance by culling the highest value at a station is the same as if that value had, by chance, not occurred, and therefore it has the smallest weighting in EVA. The larger underestimates occur when there are several culls from the upper tail and the rate of gust events is low.

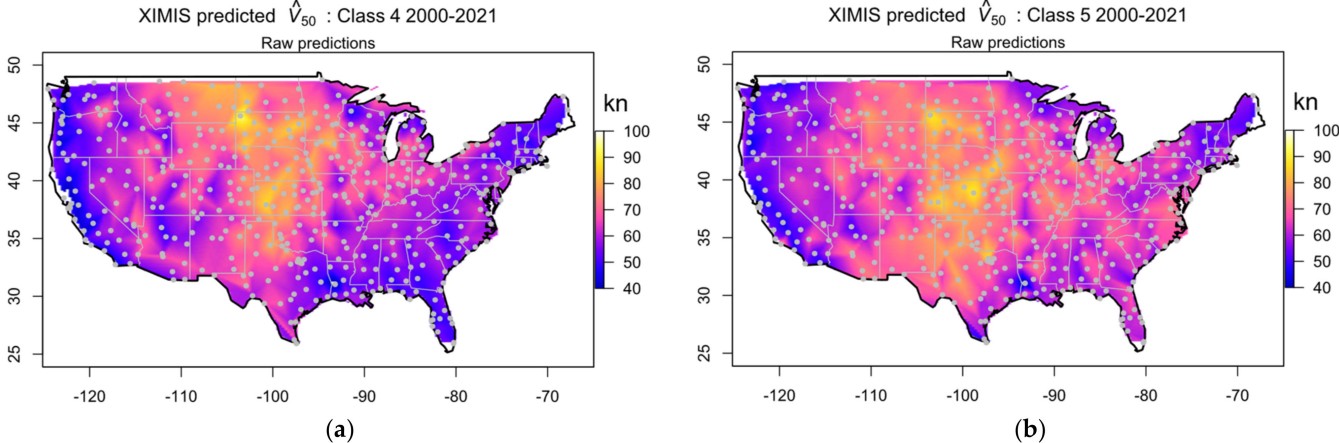

**Figure 17.** Raw XIMIS predictions of $\hat{V}_{50}$ (in kn) for: (**a**) Class 4 and (**b**) Class 5 gust events.

### 4. Discussion

For the safe design of structures, underestimation must be avoided at all costs. Overestimates are an economic issue, but Figure 14 shows that the typical effect on $\hat{V}_{50}$ is limited to about 1 kn, which is acceptable. It is the small number of stations giving larger underestimates that are the problem. Clearly, any shelter provided by obstacles at the stations must first be corrected; however, [7] shows that corrections for category 2 stations will be 10% at most for some wind directions and typically <5%. Reduction of the remaining underestimates by simple geographical smoothing is not an adequate solution because, while the underestimate at the affected station is reduced, it is redistributed over a larger surrounding area. Smoothing also reduces valid peak values. One solution is to clip the value at a station by the ensemble mean of other surrounding stations within a given radius, i.e., by taking the larger of the mean at stations within a doughnut shaped area around the station or the original station value. The density of the stations in Figure 1 clearly varies geographically, so an appropriate radius needs to be determined and might be different for each station. However, as an example, the results of using a 3° radius are shown in Figure 18, in which ~40% of stations have been clipped for each Class. The frequency of each level of underestimate correction is plotted in Figure 19, together with the names of stations where the correction is greater than 10 kn. Of these, only four are common to both Classes: Aspen, CO; Fort Polk Fullerton, LA; Harrison Boone, AR; and Hayward, WI. The

proportion of stations corrected in this way is much higher than the 10% expected from the simulation of Test 10 because it includes stations where $\hat{V}_{50}$ is legitimately lower than its neighbors and exposure corrections have not been applied. Nevertheless, it indicates that reliance on a single station without verification by its neighbors is unsafe.

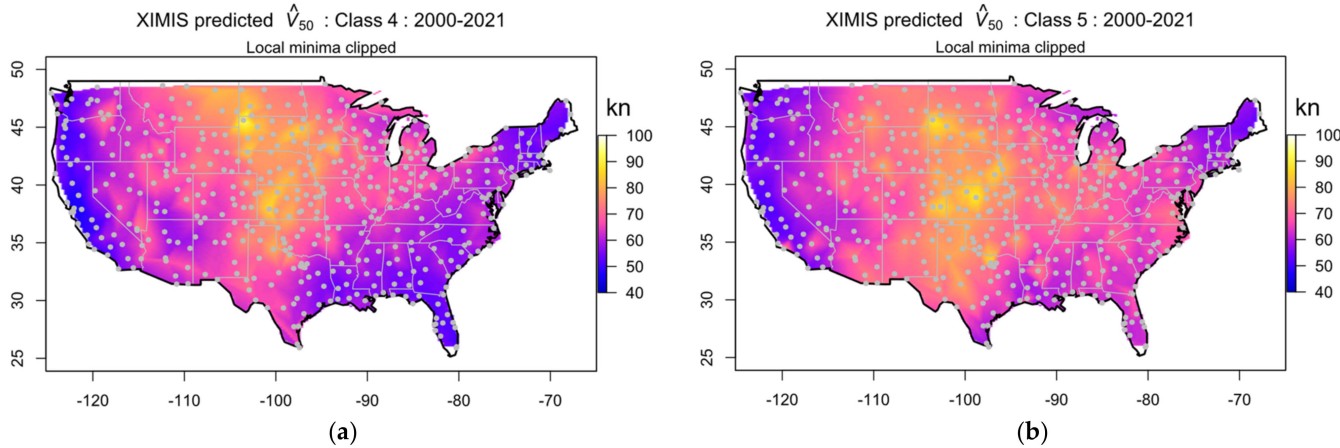

**Figure 18.** XIMIS predictions of $\hat{V}_{50}$ (in kn) with underestimates removed for: (**a**) Class 4 and (**b**) Class 5 events.

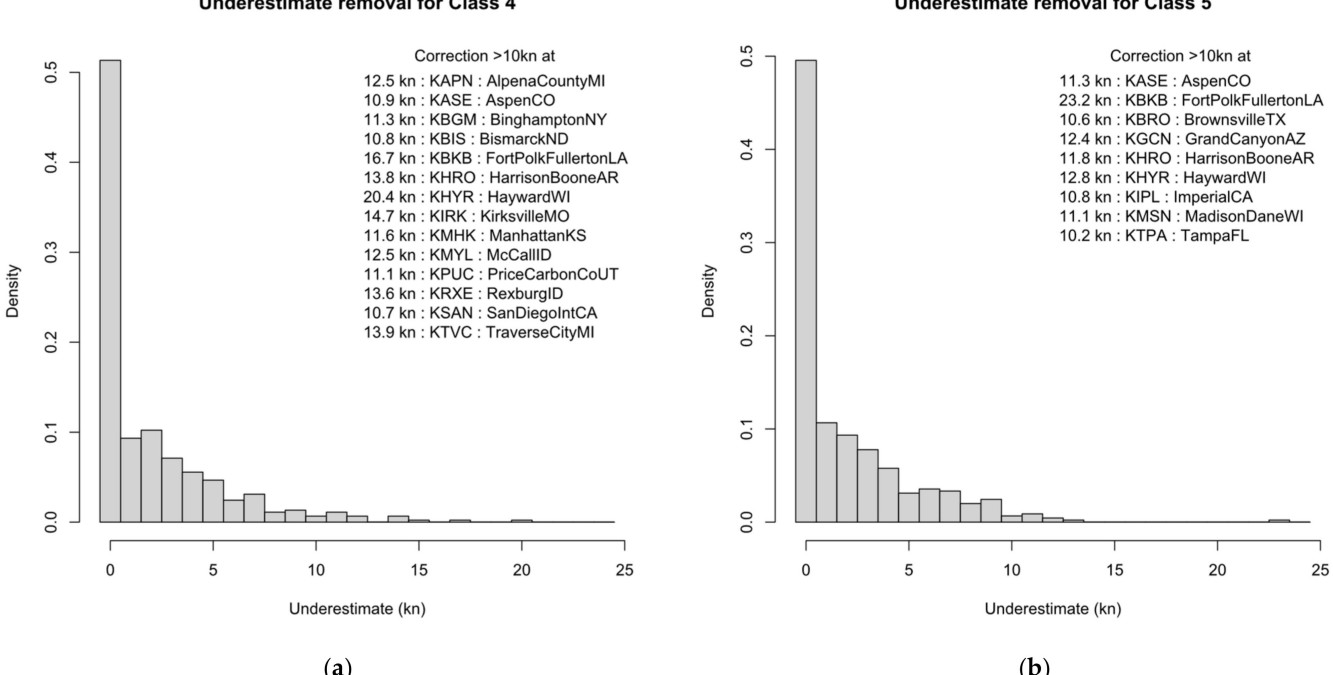

**Figure 19.** Distributions of underestimate correction of $\hat{V}_{50}$ for: (**a**) Class 4 and (**b**) Class 5 events.

The estimates of $\hat{V}_{50}$ presented in Figure 18 should be regarded as a preliminary "proof of concept" because they lack full consideration of statistical independence, asymptotic convergence, and local exposure. These considerations will be addressed in later studies and these preliminary estimates serve to inform how these studies should proceed.

As gusts from thunderstorm downbursts are the principal cause of damage, in terms of numbers and overall cost, to structures in the contiguous USA, the accurate assessment of risk posted by these events is reliant on the suitability, availability, and integrity of the relevant meteorological observations. The ASOS network provides observations at 1-min intervals, which are sufficient to resolve these mesoscale convective events, but

observations after 2013 are compromised by false-positive alerts by the introduction of a real-time quality control algorithm that culls valid data at the source stations before they are transmitted and archived. Although the number of false alerts is very small, they are biased towards these important mesoscale events. Simulation of the algorithm on observations before 2014 suggests that the peak gust is culled in about 10% of alerts for Class 4 and 5 events, and its value cannot be estimated from the surviving values. The missing data are therefore unrecoverable, but their impact on the assessment of extremes is diluted by the contributions from the unaffected events. As there is no reasonable prospect of retrospectively correcting the action of the algorithm on past observations, attention must be focused in mitigating its impact on extreme value analysis, represented here by $\hat{V}_{50}$, and the consequential risk to structures. The QC simulation predicted that the locations of underestimating stations are randomly distributed across CONUS and not predictable from the observed rates of culls. The mitigation proposed here is the removal of underestimates by ensuring that $\hat{V}_{50}$ predicted at any given station is not less than the average value of other stations within a radius (here, $3°$) around the station. This also impacts stations where $\hat{V}_{50}$ is legitimately lower than the neighbors, which represents the price that must be paid to eliminate unacceptable risk.

A principal conclusion of [2] which is reinforced by the present study is that QC Test 10, while adequate for the primary role of ASOS, is not fit for the secondary role of providing accurate observations for the assessment and regulation of the risk to buildings and structures. The conclusions of [2] have been communicated to the relevant NWS section and receipt has been acknowledged. The issue affects all ASOS wind speeds collected from 2014 and will continue to do so until some action is taken to replace Test 10 with a better algorithm.

This study completes the fourth step towards a comprehensive analysis of the risks posed by convective downbursts across CONUS. A fifth step, confirming the XIMIS [11] method as the most suitable for this context, is also completed [12]. The next step is to determine the exposure corrections for the WMO category 2 stations and to consider if it is possible to replace the worst exposed with a better-exposed nearby unused station. This is a manually intensive task, complicated by changes to the proximity and size of sheltering obstacles through time. After this, analysis of convective gust events across CONUS from 2000 to 2022 can progress.

**Funding:** This research received no external funding.

**Data Availability Statement:** The meteorological observations used in this study, TD6405 and TD6406 files from 2000 to December 2021, may be downloaded from NCEI by FTP at URL: ftp://ftp.ncdc.noaa.gov/pub/data/asos-onemin/ (accessed on 9 June 2023), or by HTTP at URL: https://www.ncei.noaa.gov/pub/data/asos-onemin/ (accessed on 9 June 2023). NCEI has lately transitioned to HTTP only, with observations from January 2022 onwards available at URL: https://www.ncei.noaa.gov/data/ (accessed on 9 June 2023), updated monthly. The R scripts and instructions to extract gust events from any ASOS station and classify by the TNNDT method in [3] are available from Mendeley at URL: https://data.mendeley.com/datasets/88jp3swkn6/1 (accessed on 9 June 2023). R scripts to replicate the analyses in this paper are available from the author, on application by email.

**Conflicts of Interest:** The author declares no conflict of interest.

## Nomenclature

The following symbols are used in this paper:

| | |
|---|---|
| $b$ | Dispersion of Gumbel (FT1) distribution |
| FT1 | Fisher-Tippett Type 1 (Gumbel) distribution |
| $m$ | Rank of observation in descending value |
| MRI | Mean Recurrence Interval (Return period) |

| | |
|---|---|
| $p$ | Probability density (PDF) |
| $P$ | Cumulative probability (CDF) |
| QC | Quality control |
| $U$ | Mode of Gumbel (FT1) distribution |
| $V$ | Wind speed |
| $\overline{V}_{2\text{min}}$ | 2-min mean wind speed |
| $\overline{V}_{3\text{s}}$ | 3-s mean wind speed |
| $\hat{V}_{3s}$ | Peak 3-s mean wind speed in previous minute |
| $\hat{V}_{50}$ | Peak $\hat{V}_{3\text{s}}$ with predicted MRI = 50 years |
| $w$ | Weibull shape parameter (in Equation (4)) |
| wLMS | Weighted least-mean-squares method |
| $y_m$ | Gumbel plotting position (linearised $\Phi$, Equation (3)) |
| $y_V$ | Gumbel reduced variate for $V$ (non-dimensional $V$, Equation (2)) |
| $\Phi$ | Annual probability of exceedance |
| $\sigma$ | Standard deviation |
| $\langle \cdot \rangle_x$ | Ensemble average of values corresponding to $x$, e.g., $\langle \overline{y}_m \rangle_{\hat{V}}$ |

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
