# Peer review of "Impact of ASOS Real-Time Quality Control on Convective Gust Extremes in the USA"

_2674-0494, doi:10.3390/meteorology2020017_

Round 1
Reviewer 2 Report
The manuscript discusses the impact of the erroneously culled observations by the real-time quality control algorithm of the US Automated Surface Observation System on the estimates of 50-year mean recurrence interval gust speeds. The topic is interesting and worth investigating. The paper is logically sound and well written. Several minor suggestions are given below for the author to consider:
- There are some formatting issues. The cross references of figures are incorrectly displayed as “Error! Reference source not found.” I believe this issue was caused by using the journal’s template and can be easily addressed by the author.
- There are many different “classes” in this paper, including L39 “WMO class 1 and 2”, “six characteristic Classes”, and L95 “5 event classes”. Adding a simple introduction to these classes would improve the readability of the paper.
- The quality of some figures could be further improved, e.g., Figures 6-8&16-18, the outline of the country is not fully filled in with the heatmap, and the unit of the colorbar (e.g., kn) could be added.
- In addition to the papers by the author, it is suggested to review more relevant papers in the introduction.
- L73, it is suggested to show how kn can be converted to km/h or m/s.
- L374, is there any special reason for using a 3° radius for clipping, why not use 1°, 2°, 4°, 5°, has the author conducted any sensitivity test on it?
Author Response
- There are some formatting issues. The cross references of figures are incorrectly displayed as “Error! Reference source not found.” I believe this issue was caused by using the journal’s template and can be easily addressed by the author. >> I am familiar with this form of error which is why the automatic cross-referencing features of Word have not been used. All figures are manually cross referenced in the text. I can find no instances of this error in my copies.
- There are many different “classes” in this paper, including L39 “WMO class 1 and 2”, “six characteristic Classes”, and L95 “5 event classes”. Adding a simple introduction to these classes would improve the readability of the paper. >> Yes. Now “WMO exposure categories”, so that Classes is used only in the event classification. These are now listed in the Introduction.
- The quality of some figures could be further improved, e.g., Figures 6-8&16-18, the outline of the country is not fully filled in with the heatmap, and the unit of the colorbar (e.g., kn) could be added. >>Values are interpolated within the envelope of the station locations and masked outside the CONUS boundary. However, they are not extrapolated outside the envelope, which is why the tip of Florida is not filled. Units added to key.
- In addition to the papers by the author, it is suggested to review more relevant papers in the introduction. >> This study is focused on a specific problem. Relevant references are given in the extended introduction. No-one else has contributed usefully to this issue.
- L73, it is suggested to show how kn can be converted to km/h or m/s. >> Yes.
- L374, is there any special reason for using a 3° radius for clipping, why not use 1°, 2°, 4°, 5°, has the author conducted any sensitivity test on it? >> No and no. Please see the revised text.
Reviewer 3 Report
In my opinion, the paper under review is a very interesting and relevant study of great practical interest. From a scientific point of view, there are no objections to the correctness of the problem statement and analysis of the results. The results obtained by the author are novel, and their analysis is quite correct. On the merits of the article, I have no comments. But I would like to invite the author to make some structural additions and changes to the article that would help improve it. The author needs to clearly formulate the purpose of the work in the introduction, but first discuss the problem on the basis of available publications. The author needs to state:
- scientific novelty;
- practical significance;
- research objectives.
In conclusion, it is advisable for the author to discuss what has been achieved and what contribution (scientific and practical) this article has. It would also be nice to outline the prospects for further research.
Author Response
Review 3
In my opinion, the paper under review is a very interesting and relevant study of great practical interest. From a scientific point of view, there are no objections to the correctness of the problem statement and analysis of the results. The results obtained by the author are novel, and their analysis is quite correct. On the merits of the article, I have no comments. But I would like to invite the author to make some structural additions and changes to the article that would help improve it. The author needs to clearly formulate the purpose of the work in the introduction, but first discuss the problem on the basis of available publications. The author needs to state:
- scientific novelty;
- practical significance;
- research objectives.
>> See revised and extended Introduction and Discussion.
In conclusion, it is advisable for the author to discuss what has been achieved and what contribution (scientific and practical) this article has. It would also be nice to outline the prospects for further research.
>> Yes, see final paragraphs of Discussion.
Round 2
Reviewer 1 Report
The author has adequately addressed all of my comments. The description of methods and presentation of results are much clearer. Also, the structural changes and additions to the introduction make the paper easier to read. Any changes that were not addressed in the second draft were well justified and explained in the cover letter. This is a high-quality manuscript and I recommend it be accepted in its present form.